# Selective Laser Sintering Fabricated Thermoplastic Polyurethane/Graphene Cellular Structures with Tailorable Properties and High Strain Sensitivity

**Alfredo Ronca [1,†], Gennaro Rollo [2,†], Pierfrancesco Cerruti [2], Guoxia Fei [3], Xinpeng Gan [3], Giovanna G. Buonocore [4], Marino Lavorgna [4,*]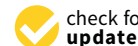, Hesheng Xia [3,4,*], Clara Silvestre [2] and Luigi Ambrosio [1]**

[1] Institute of Polymers, Composites and Biomaterials, National Research Council Viale J.F. Kennedy, 54-80125 Naples (Na), Italy; alfredo.ronca@cnr.it (A.R.); luigi.ambrosio@cnr.it (L.A.)

[2] Institute of Polymers, Composites and Biomaterials, National Research Council, Via Campi Flegrei, 34 80078 Pozzuoli (Na), Italy; gennaro.rollo@ipcb.cnr.it (G.R.); cerruti@unina.it (P.C.); clara.silvestre@ipcb.cnr.it (C.S.)

[3] State Key Laboratory of Polymer Materials Engineering, Polymer Research Institute, Sichuan University, Chengdu 610065, China; feiguoxia1981@163.com (G.F.); xinpenggan@163.com (X.G.)

[4] Institute of Polymers, Composites and Biomaterials, National Research Council, P. le Enrico Fermi, 1-80055 Portici (Na), Italy; gbuonoco@unina.it

**\*** Correspondence: mlavorgn@unina.it (M.L.); xiahs@scu.edu.cn (H.X.); Tel.: +39-081-7758838 (M.L.); +86-28-85460535 (H.X.)

† These authors contributed equally to this work.

**Abstract:** Electrically conductive and flexible thermoplastic polyurethane/graphene (TPU/GE) porous structures were successfully fabricated by selective laser sintering (SLS) technique starting from graphene (GE)-wrapped thermoplastic polyurethane (TPU) powders. Several 3D mathematically defined architectures, with porosities from 20% to 80%, were designed by using triply periodic minimal surfaces (TMPS) equations corresponding to Schwarz (S), Diamond (D), and Gyroid (G) unit cells. The resulting three-dimensional porous structures exhibit an effective conductive network due to the segregation of graphene nanoplatelets previously assembled onto the TPU powder surface. GE nanoplatelets improve the thermal stability of the TPU matrix, also increasing its glass transition temperature. Moreover, the porous structures realized by S geometry display higher elastic modulus values in comparison to D and G-based structures. Upon cyclic compression tests, all porous structures exhibit a robust negative piezoresistive behavior, regardless of their porosity and geometry, with outstanding strain sensitivity. Gauge factor (GF) values of 12.4 at 8% strain are achieved for S structures at 40 and 60% porosity, and GF values up to 60 are obtained for deformation extents lower than 5%. Thermal conductivity of the TPU/GE structures significantly decreases with increasing porosity, while the effect of the structure architecture is less relevant. The TPU/GE porous structures herein reported hold great potential as flexible, highly sensitive, and stable strain sensors in wearable or implantable devices, as well as dielectric elastomer actuators.

**Keywords:** selective laser sintering (SLS); thermoplastic polyurethane (TPU); graphene (GE); mathematically defined structures; piezoresistivity; strain sensors

## 1. Introduction

Additive manufacturing, also known as 3D printing (3DP), is an innovative manufacturing technology which allows one to turn complex 3D models into real objects without special tooling and

with extreme facility, cost, and time savings, alongside high accuracy in the realization of specific complex items [1,2]. Additionally, 3DP includes several technologies, such as stereolithography (SLA) [3], fused deposition modeling (FDM) [4], and selective laser sintering (SLS), and some less common techniques [5–7]. Among them, SLS ensures the highest geometrical freedom and dimensional precision, which allows the manufacturing of parts with well-defined prototypes and components applied in different fields, including electronics, mechanics, and biomedicine [1,8]. Starting from a computer-aided design (CAD) 3D model, SLS builds up objects by sintering and fusing powder material in a layer-by-layer approach, via a computer-controlled laser [9,10]. Generally, thermoplastic polymers are mostly used for the laser sintering process [11]. However, only a few polymers are now commercially available, with polyamides (PA-11 and PA-12) being the most used, while polystyrene [12], polycarbonate [13], thermoplastic polyurethane (TPU) [14], and their composites are seldom explored or used in specific sectors. Despite the continuous progress in the optimization of SLS technology, many critical issues still remain unsolved, including the possibility to manufacture multifunctional conductive parts able to exhibit both electrical conductivity alongside lightweight and elastic properties. In this context, it is very interesting to develop new powders made up of conductive nanoparticles dispersed in or coated onto elastomeric particles [15,16]. Piezoresistive structures, realized by using conductive elastomeric polymers, are commonly used for load/pressure sensors and actuators due to their quick response to external stress [17]. In these systems, the mechanical deformation of the structure brings about a change in the conductive pathway by modifying the mean particle distance between the conductive nanoparticles, and therefore the material's resistivity [18]. Several fillers able to realize a 3D interconnected conductive network, such as carbon black (CB) [19], carbon fiber (CF) [20], carbon nanotubes (CNT) [21,22], and graphene (GE) [23,24] have been used to modify the polymer matrix and realize conductive composites endowed with advanced functional properties, including chemical sensing, capacitance, and piezoresistivity. Among them, GE has attracted huge interest because of its excellent conductivity (3000–5000 W/m·K), high carrier mobility ($\approx$10.000 cm$^2$/V·s), optical transparency ($\approx$97.7%), and high Young's modulus ($\approx$1 TPa) [25,26]. Thus, several recent studies have focused on the electrical and thermal conductivity of GE-based composites by using polydimethylsiloxane (PDMS) [27], polyamide-6 [28], and natural rubber [29] as the matrix. Among the elastomeric matrices, thermoplastic polyurethane (TPU) is a versatile polymer, as its morphology is made up of a hard diisocyanate segment, and a soft segment [30,31] consisting of a tailor-designed alkyl, polyether, or polyester chain. Due to its peculiar morphology, TPU exhibits unique thermomechanical properties and strong capability of shape recovery upon loading/unloading cycles [32]. Xia et al. already demonstrated the feasibility of SLS technique to construct compact 3D electrically conductive materials by processing TPU powder wrapped with CNTs [1]. They substantiate the selection of CNTs as the best filler to allow better coalescence of powders during the laser sintering process, in order to maximize the mechanical properties. Moreover, the possibility of using TPU powders wrapped with 2D filler as GE, which hinders, to some extent, the coalescence of particles during the sintering, allows for tailoring of the structural and functional properties of the resulting porous structures in a value-range not yet explored. The control of pore morphology and dimension, which depends on the shape of the unit cell, from which the three-dimensional structure is generated, affects their mechanical and electrical response, making them more sensitive to mechanical stress/strain, thus enhancing their stress sensor capability. Herein, the effect of pore morphology and distribution on the thermal, mechanical, and piezoresistive properties of porous structures fabricated by SLS technique by using a home-made powder consisting of TPU wrapped with GE platelets (TPU/GE porous structures) is investigated. Three-dimensional, mathematically-defined architectures have been designed and realized starting from triply periodic minimal surface (TMPS) geometry. More specifically, three different geometries have been used, namely Gyroid [33], Diamond [34], and Schwarz [35], with an extent of porosity ranging from 20% to 80%. Electrical and thermal conductivity, mechanical strength, filler dispersion, and interaction with the polymer matrix of the TPU/GE porous structures are investigated and correlated with their porosity and morphology. The reported results are of

interest for the design and fabrication of novel 3D printable strain sensors, as well as lightweight thermal conductors.

## 2. Materials and Methods

A polyester-type thermoplastic polyurethane (TPU) was used as the matrix phase (LUVOSINT X92A-1 – Lehmann & Voss, Hamburg, Germany, 2016). Graphene material was provided by Deyang Carbonene Technology Co., Ltd., Deyang, China, 2016. Silica nanoparticles, mainly used to promote the flowing of TPU particles, consists of fine powder with particle size less than 10 nm, and it was purchased from Nanjing Tianxing New Material Co., Ltd., Nanjing, China, 2016. All of the materials and reagents were used as received.

### 2.1. Preparation of Thermoplastic Polyurethane/Graphene (TPU/GE) Nanocomposites Powder

The method of preparing composite powder for SLS is of great importance, as it directly determines the dispersion of nanofiller in the polymer matrix, also affecting the properties of the SLS fabricated porous structures. GE was dispersed in ethanol and subjected to ultrasonication for 12 h to get a homogenous dispersion. The TPU powders were then added to the GE suspension and mechanically stirred for 4 h. Then the mixture was filtered with a Buchner funnel under reduced pressure. The obtained GE-coated TPU powders were dried in a vacuum oven at 50 °C for 24 h. Subsequently, the TPU/GE powders were sieved to remove particles with size over 40μm. In addition, 0.2 wt% silica was used to further improve the powder flowing ability.

### 2.2. Porous Structures Design by Triply Periodic Minimal Surfaces (TPMS)

To design 3D porous structures, a mathematical approach has been used, starting from triply periodic minimal surface (TPMS) equations. TPMS are minimal surfaces periodic in three independent directions, extending infinitely, and in the absence of self-intersections, partitioning the space into two labyrinths. Wolfram Mathematica software was used to generate CAD-files that describe the surfaces of Gyroid (G), Diamond (D), and Schwarz (S) architectures at different porosity. The following trigonometric equations were used with boundary condition x, y, z = $[-3\pi; 3\pi]$:

$$G: \ \sin(y) + \cos(y) \cdot \sin(z) + \cos(z) \cdot \sin(x) = C \tag{1}$$

$$D: \ \sin(x) \cdot \sin(y) \cdot \sin(z) + \sin(x) \cdot \cos(y) \cdot \cos(z) + \cos(x) \cdot \sin(y) \cdot \\ \cos(z) + \cos(x) \cdot \cos(y) \cdot \sin(z) = C \tag{2}$$

$$S: \ G \cos(x) + \cos(y) + \cos(z) = C \tag{3}$$

In these equations, the C parameter is the offset which controls the porosity of the structures. An accurate study has been conducted in order to understand the correlation between percentage of porosity and offset value C and it has been reported in Section 3.1. Three different porosity values (40%, 60%, and 80%) have been set for each geometry in order to study the effect of porosity on thermal and electrical conductivity. Rhinoceros software was used to scale the CAD-files to the required dimensions in order to obtain a $10 \times 10 \times 10$ mm$^3$ sized cube. In the following, Gyroid, Diamond, and Schwarz-based porous structures are labelled as GX, DX, and SX, respectively, where G, D, or S represents the geometry and X represents the % of porosity. As an example, G20 stands for Gyroid-based architecture with 20% porosity.

### 2.3. Nanocomposite Porous Structure Realization by Selective Laser Sintering (SLS)

A HT251P SLS Equipment (Farsoon Hi-tech, Changsha, China, 2015) was used as 3D printer. The SLS procedure is described briefly as follows: graphene wrapped TPU powder was spread out on the sample tray and preheated at 60 °C, N$_2$ was used in the chamber as purging gas. The laser

selectively fused the powder based on the CAD model, according to the processing parameters reported in Table 1.

**Table 1.** Sintering parameters adopted for processing thermoplastic polyurethane/graphene (TPU/GE) composite powders.

| Process Parameters | Value |
| --- | --- |
| Laser power (W) | 60 |
| Laser scan spacing (μm) | 100 |
| Laser scan speed (m/s) | 7.6 |
| Part bed temperature (°C) | 95 |
| Powder feed temperature (°C) | 65 |
| Outline laser power (W) | 5 |
| Layer thickness (μm) | 150 |

An outline laser power of 5 W has been used in order to prevent the sample from sticking to the powder and causing a decrease in accuracy. After processing, the porous specimens were allowed to cool inside the equipment chamber for approximately 1 h and then they were removed from the printer and sprayed with compressed air to remove non-sintered powder from the interstices and porosity.

### 2.4. Scanning Electron Microscopy (SEM)

The porous morphology of the several printed specimens was studied by scanning electron microscopy (SEM) by using a FEI Quanta 200 FEG-SEM microscope (FEI Company, Hillsboro, OR, USA, 2009). The samples were fixed on a support and metallized with a gold-palladium alloy to ensure better conductivity and prevent the formation of electrostatic charges.

### 2.5. Transmission Electron Microscopy (TEM)

Transmission electron microscopy (TEM) imaging was performed by using a Tecnai G2 Spirit TWIN electron microscope (FEI, FEI Company, Hillsboro, OR, USA, 2009) operating at 120 kV on 100 μm TEM cryosections.

### 2.6. Thermal Properties

Thermal properties of TPU and TPU/GE were measured by differential scanning calorimetry (DSC) and thermogravimetric analysis (TGA). The DSC measurements were performed with a TA Instrument DSC Q2000. Samples of 5 mg were heated up to 250 °C at a heating rate of 10 °C/min, then cooled to −50 °C at 10 °C/min and reheated to 250 °C at 10 °C/min under nitrogen atmosphere.

Thermogravimetric analysis was carried out on approximately 8 mg samples by using a PerkinElmer Pyris Diamond TG/DTA. The samples were pre-heated to 90 °C at 10 °C/min for 10 min, then subject to a ramp up to 800 °C at a heating rate of 5 °C/min under nitrogen atmosphere.

### 2.7. Raman Spectra Analysis

Raman spectra were obtained using a Horiba Jobin Yvon LabRam ARAMIS model, with a 532 nm laser (green light), hole 300 μm, slit 300 μm, objective ×50/0.50, grating 600, time 10 s.

### 2.8. Mechanical and Piezoresistive Measurements

Static compression tests were carried out by using a mechanical testing machine (Instron 5564 dynamometer, Norwood, MA, USA, 1997) and the $10 \times 10 \times 10$ mm$^3$ cubic specimens were compressed at a strain rate of 3 mm/min. Electrical and compression tests were carried out simultaneously to evaluate the piezoresistive properties of the 3D printed structures. Thus, coupled to the mechanical testing machine a multimeter (Keysight 34401A $6\frac{1}{2}$ Digit Multimeter, Agilent Technologies, Santa Clara, CA, USA, 2006), which was controlled by a homemade LabVIEW program, was used to measure the

change of electrical resistance with the applied load and induced deformation. Two electrodes, made of copper conductive tape, were glued on the top and on the bottom of the specimen and connected with the multimeter through copper wires. The mechanical properties were evaluated by submitting the samples to a cyclic compressive strain/unstrain up to 8% of initial value of the length of cubic sample, with a deformation rate of 3 mm/min, at 25 °C. Before measurement, the porous structures were pre-compressed to a strain value of 4%. The electrical resistance of the specimen was monitored simultaneously to compression testing. The strain sensitivity of the samples was expressed as Gauge Factor, $GF = (\Delta R/R_0 \cdot \varepsilon)$, where $\Delta R/R_0$ is the resistance change rate and $\varepsilon$ is the compression strain.

## 2.9. Thermal Conductivity Measurement

Porous cylindrical specimens, characterized by a height of 4 mm and a diameter of 21 mm, were realized by SLS with the three proposed geometries (G, D, and S) and used for thermal conductivity measurements. The thermal conductivity ($\lambda$) was measured by a thermal analyzer (TPS2500, Hot Disk, Göteborg, Sweden, 2010).

## 3. Results and Discussion

### 3.1. Design and Realization of the TPU/GE Porous Structures

TPU/GE composite powder was used in the SLS printing process in order to build three porous structures by using Schwarz, Diamond, and Gyroid unit cells. Moreover, an accurate study to understand the correlation between the offset factor (C) present in the Equations (1)–(3), and the structure porosity of the structure is reported in Figure 1. The porosity linearly decreases by increasing the offset value, and this enables design of structures in a range of porosities (from 40% to 80%).

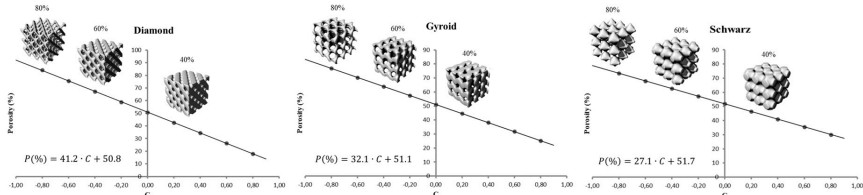

**Figure 1.** Correlation between percentage of porosity and C value for Diamond, Gyroid and Schwarz-based unit cells architectures.

Porous structures, consisting of 3 × 3 × 3 unit cells, were SLS printed by using the TPU/GE composite powder with three different porosity values. Figure 2 clearly shows that the three different unit cells give rise to a different distribution of pores within the resulting 3D structure. In particular, the Schwarz unit cells bring about a structure with bigger pores, and so the trabeculae between pores (i.e., struts in the foams) are bigger. The structures generated by G and D unit cells present more pores with smaller dimensions, and consequently the trabeculae have a smaller size.

The designed architectural features are preserved and porosity is almost unaffected by the fabrication process, as shown in Figure 2. These results clearly show the suitability of the TPU/GE powder to print porous structures with narrow pore size distributions and high pore interconnectivity by SLS manufacturing.

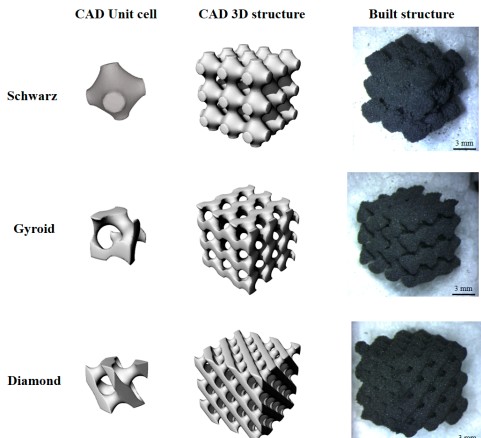

**Figure 2.** Visualization of the three designed porous structures. CAD-designs of the unit cells (left column); CAD-designs of 3 × 3 × 3 assembled structures (center column); photos of the TPU/GE SLS fabricated structures (right column).

### 3.2. Chemical-Physical and Morphological Characterization of the SLS Manufactured Foams

SLS processing involves the selective melting of the particle surface by using a laser beam. In this process, the TPU particles coalesce with each other, building up the desired 3D structure [1]. Since the TPU melt is highly viscous and no stress is generated during the process, the particle morphology is not significantly changed. Therefore, similar to the CNTs [1], the GE sheets remain entrapped in between the particle boundaries, thereby forming a percolated conductive network, as sketched in Figure 3a. Low magnification SEM images (Figure 3b) show that the wall structure of the holes in the porous specimens consist of sintered TPU particles. High magnification images (Figure 3c) clearly demonstrate that the surface of the TPU particles is covered by GE platelets. TEM observations provide additional information on the morphology and microstructure of the samples. Figure 3d and 3e show the GE percolated network due to the filler segregation between the sintered TPU particles, with a thickness ranging from 200 to 500 nm.

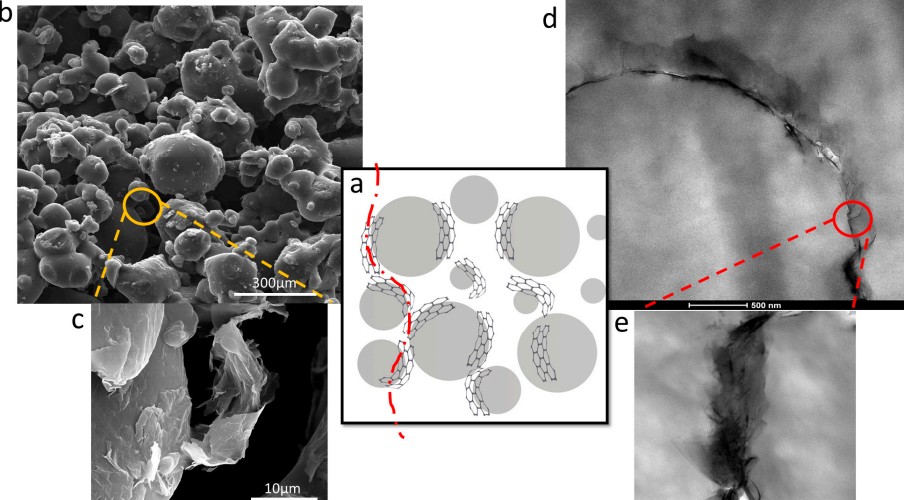

**Figure 3.** (**a**) Schematics of the microstructure of the fabricated porous TPU/GE composites (GE content 1.0 wt%), highlighting the percolated GE network at the interparticle boundary (red dashed line). SEM images of the (**b**) wall structure of the composite, and the (**c**) GE-wrapped TPU particle surface. (**d**,**e**) TEM images of the percolated GE network in the SLS-processed composites.

Raman spectroscopy was used to get insight on the effect of SLS processing on the structure of the GE platelets within the sintered porous structures. The analysis has been conducted by using a

Raman microscopy, in order to selectively focus on the TPU particles or onto the segregated graphene between particles. Raman spectra of TPU, GE powder before processing, and GE in the fabricated porous structures are reported in Figure 4. The spectrum of pristine TPU shows the typical peaks of polyurethane, including the absorption peak of aromatic rings (1470–1440 cm$^{-1}$), the absorption peak at 1665 cm$^{-1}$ corresponding to the C=C stretching, the C=O bending peak of the ester group at 1740 cm$^{-1}$, and the peak at about 3000 cm$^{-1}$ due to C-H bonds [36]. In the Raman spectrum of graphene, it is possible to observe the G (1580 cm$^{-1}$) band, which is a primary in-plane vibrational mode of carbon-carbon bonds in graphene sheets, the D (1350 cm$^{-1}$) band ascribed to disordered carbon in graphene and 2D (2690 cm$^{-1}$) band, which is a second-order overtone of D band ascribed to AB-stacked graphene (where AB-Stacked refers to misorientation of graphene nanoplatelets) [37]. Figure 4 demonstrates that no dramatic change occurs in the GE spectrum when graphene nanoplatelets are assembled onto the TPU particle surface and then sintered during SLS processing. In any case, a slight decrease of I$_D$/I$_G$ ratio is observed, which results in 0.064 for GE powder and 0.035 for the GE in the composite realized by SLS. This variation may be tentatively ascribed to an effect of the sintering process, which likely reduces the extent of defects of GE platelets assembled onto TPU powder particles (the effect of TPU in the measurements of the I$_D$/I$_G$ ratio is negligible) [38,39].

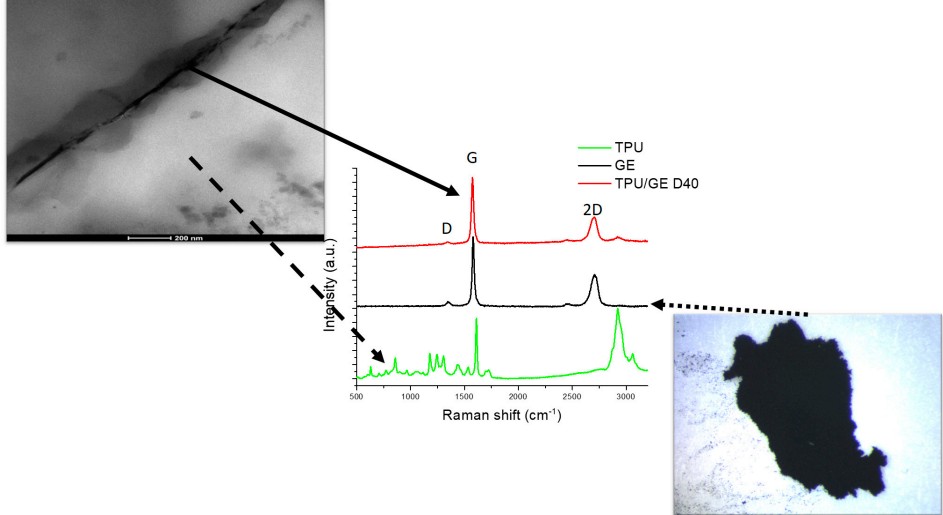

**Figure 4.** Raman spectra of GE powder (black line), and TPU (green line) and GE (red line) in the TPU/GE G40 composite after SLS processing.

Thermal Properties

Thermal properties of SLS-fabricated TPU-based and TPU/GE porous samples were investigated by DSC and TGA. Figure 5a shows the cooling and heating DSC curves of the reference TPU porous structure. From both curves, a main thermal event is noticed, consisting of a first-order transition showing large thermal hysteresis, as indicated by the peak maximum recorded at 91.2 ± 0.3 °C and 166 ± 1.2 °C upon cooling and heating, respectively. This transition is related to the melt crystallization and fusion of TPU hard segment crystallites [40]. The calculated melting enthalpy value was as low as 5.4 ± 0.4 J/g, indicating that only a small fraction of the material was able to crystallize [41]. In the heating thermogram, it is also worth noting the occurrence of the glass transition temperature ($T_g$) at −18 °C.

Figure 5a also reports the DSC thermogram of TPU/GE as representative of the thermal behavior of the SLS processed TPU/GE composite materials. Crystallization and melting peaks were detected at 65.47 ± 0.40 °C and 144.65 ± 1.01 °C, respectively, indicating that the addition of GE hindered TPU crystallization, also decreasing the crystalline size of the TPU fraction [40,42]. In addition, the value of melting enthalpy was about 4.9 ± 0.3 J/g, showing that an even smaller amount of hard segments

crystallized in comparison with the plain TPU. Finally, GE also caused a significant increase in $T_g$, which was detected at $-11$ °C in the composites. The rise in $T_g$ shows that the presence of the carbonaceous filler was able to mediate the H-bonding interactions between TPU chains, reducing the mobility of the polymer soft segments, as already reported for graphene/TPU composites [43]. Similar results were obtained for all porous systems regardless of geometry and porosity.

GE also affected the thermal stability of the TPU foam. In Figure 5b, the thermogravimetric curves of TPU and TPU/GE are compared. TPU degradation occurs with a two-step mechanism. The first process, attributed to the cleavage of urethane bonds of TPU [44], starts at about 280 °C, with a maximum rate at 308.30 ± 1.20 °C, and accounts for about 30% mass loss. The second weight loss step, related to the decomposition of soft segments of TPU, had a maximum rate at 388.12 ± 1.23 °C, leading to a residual char value of 1.2%. The presence of 1 wt% GE significantly retarded the degradation onset, which occurred at about 301.22 ± 3.37 °C, also shifting the degradation rate maximum at 342.74 ± 2.32 °C. Therefore, the addition of GE brings about an improvement of thermal stability of TPU, as the large-area graphene sheets increase the tortuous path for the volatile products to be released, also resulting in a higher amount of residual char (8.5%) [45].

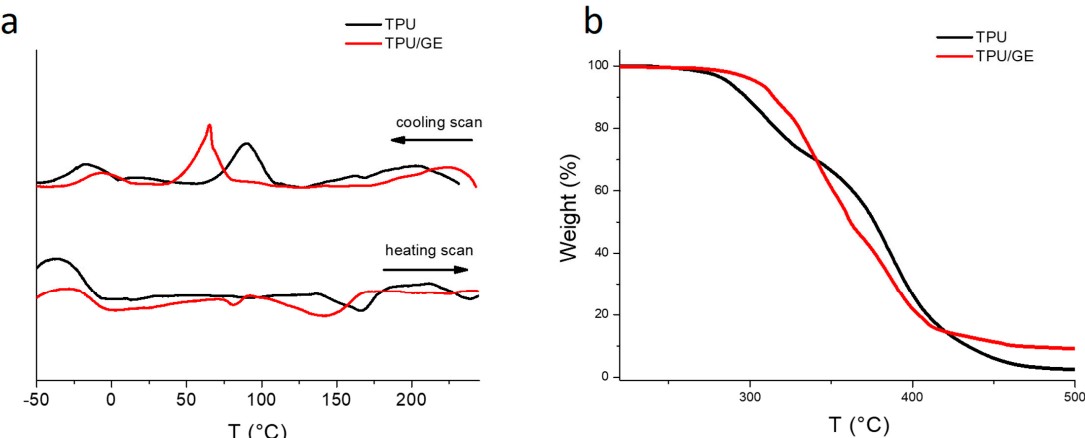

**Figure 5.** (**a**) DSC and (**b**) TGA curves of SLS fabricated TPU and TPU/GE nanocomposite.

### 3.3. Mechanical and Piezoresistive Characterization

The effect of porosity and geometry on the mechanical behavior of the TPU/GE porous structures was investigated by compression tests. Figure 6a shows the stress-strain curves for all investigated samples. An initial toe region caused by a take-up of slack and alignment of the specimen followed by a linear region can be observed. Elastic modulus was calculated from the linear region of the curves for all the geometries considered (Figure 6b). In particular, the samples were tested at small strain values (<10%), in order to ensure that all samples were in their elastic deformation region. D and G architecture structures show, in the deformation range which has been investigated, a linear increase of stress with increasing strain, while the S geometry structures exhibit a progressive strengthening during compression, which results in higher stress values in comparison to the corresponding D and G structures.

This outcome is ascribed to the different morphology of pores present in the systems obtained by starting from S unit cells with respect to D and G systems. In fact, the structures with S geometry result in having less pores with bigger dimensions [33–35]. This implies that at a given porosity value, the average thickness of the trabeculae in the S structure is bigger, so the mechanical stress required to get a defined deformation is larger (as compared to other structures with same porosity). The porous structures exhibit a dramatic enhancement of the elastic compression modulus (more than 2 orders of magnitude in the case of D and G structures) when the porosity decreases from 80% to 40%. Moreover, unit cells (i.e., S, D, or G) also affected the mechanical performance of the porous structures, with the S structures being significantly stiffer than the corresponding G and D-based structures.

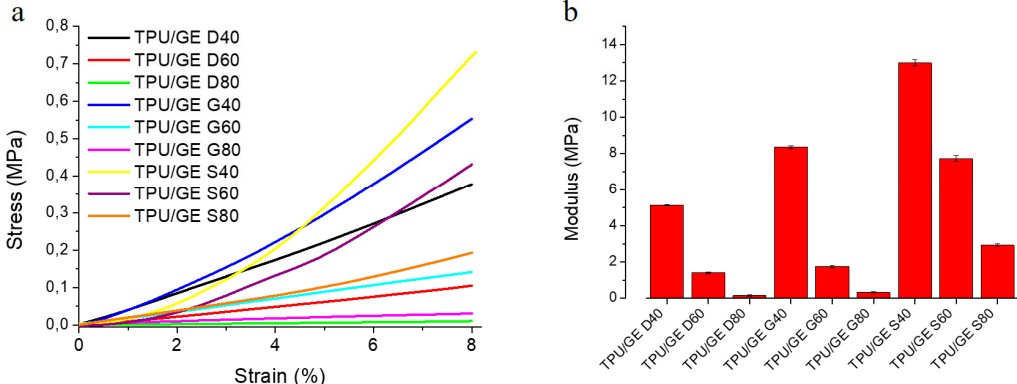

**Figure 6.** (**a**) Compression stress-strain curves, and (**b**) compression elastic modulus of the SLS fabricated TPU/GE porous structures.

All SLS fabricated structures, except for TPU/GE D80, which gave no reliable results, were tested as concerning their electrical conductivity. Indeed, in the presence of graphene, the insulating polymer matrix became conductive, due to the formation of a segregated percolated graphene network at the boundary of the TPU particles. While neat TPU displayed a conductivity value of $10^{-13}$ S/m, all the porous structures exhibited values ranging from $7 \times 10^{-5}$ (TPU/GE S80) to $9 \times 10^{-4}$ S/m (TPU/GE G80). These conductivities are comparable with those reported in literature for graphene/TPU foams fabricated by thermal induced phase separation [41,42], indicating the formation of a stable graphene conductive network in all samples.

The piezoresistive behavior of the porous TPU/GE porous structures was studied by submitting the samples to compression cycles with strain up to 8%. Figure 7 shows the results characterizing the piezoresistive behavior of the D, G, and S-based structures with 40% porosity. All samples showed a negative piezoresistive behavior that is the electrical resistance decreasing with increasing strain.

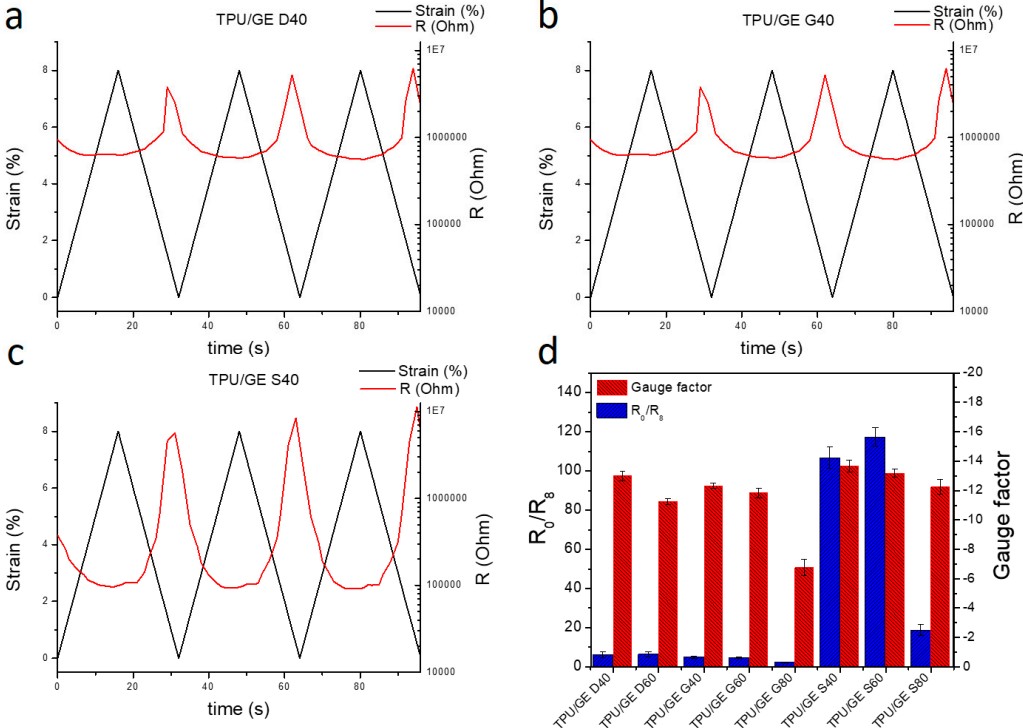

**Figure 7.** Piezoresistive behavior of (**a**) TPU/GE D40, (**b**) TPU/GE G40, and (**c**) TPU/GE S40 under cyclic compression. (**d**) Resistance ratio and gauge factor of TPU/GE porous structures at 8% compression strain.

This outcome arises from the compactness of TPU particles and the enhancement of the GE nanoplatelet contact upon compression, which leads to the formation of more conductive pathways [46]. Electrical conductivity strongly depends on the percentage of porosity but also on internal architectures of the 3D structures, both at low (i.e., zero deformation corresponding to the case of constant preload applied to the samples) and high deformation (i.e., 8% deformation in compression). In detail, the conductivity at zero deformation decreases as the percentage of porosity increases. In fact, considering the S geometry, σ decreased by about one order of magnitude when the porosity increased from 40% to 80%, going from $1.17 \times 10^{-5}$ to $1.50 \times 10^{-6}$. Similar results have been obtained when the deformation is 8%. Moreover, the piezoelectrical sensitivity varies significantly with the three geometries, as it is possible to see from the ratio of electrical resistance ($R_0/R_8$), whose values are reported in Table 2.

**Table 2.** Electrical conductivity of TPU/GE composites as a function of porosity and internal morphology.

| Sample | σ (S/m) | | ρ (Ω·m) | | $R_0/R_8$ |
|---|---|---|---|---|---|
| | 0% | 8% | 0% | 8% | |
| D40 | $3.49 \times 10^{-5} \pm 8.48 \times 10^{-6}$ | $1.86 \times 10^{-4} \pm 5.59 \times 10^{-6}$ | $3.03 \times 10^{4} \pm 7.89 \times 10^{3}$ | $5.38 \times 10^{3} \pm 1.67 \times 10^{2}$ | $6.12 \pm 1.65$ |
| D60 | $5.01 \times 10^{-6} \pm 1.77 \times 10^{-6}$ | $2.72 \times 10^{-5} \pm 1.16 \times 10^{-6}$ | $2.14 \times 10^{5} \pm 4.83 \times 10^{4}$ | $3.69 \times 10^{4} \pm 2.83 \times 10^{3}$ | $6.33 \pm 1.41$ |
| D80 | N.A. | N.A. | N.A. | N.A. | N.A. |
| G40 | $3.00 \times 10^{-5} \pm 3.42 \times 10^{-6}$ | $1.80 \times 10^{-4} \pm 6.58 \times 10^{-6}$ | $2.52 \times 10^{4} \pm 2.32 \times 10^{3}$ | $5.55 \times 10^{3} \pm 2.02 \times 10^{2}$ | $4.95 \pm 0.50$ |
| G60 | $1.76 \times 10^{-5} \pm 9.25 \times 10^{-7}$ | $7.51 \times 10^{-5} \pm 4.80 \times 10^{-6}$ | $5.70 \times 10^{4} \pm 3.08 \times 10^{3}$ | $1.34 \times 10^{4} \pm 9.59 \times 10^{2}$ | $4.66 \pm 0.46$ |
| G80 | $1.32 \times 10^{-5} \pm 3.67 \times 10^{-7}$ | $2.68 \times 10^{-5} \pm 2.06 \times 10^{-6}$ | $7.60 \times 10^{4} \pm 2.13 \times 10^{3}$ | $3.75 \times 10^{4} \pm 3.07 \times 10^{3}$ | $2.21 \pm 0.20$ |
| S40 | $1.17 \times 10^{-5} \pm 5.67 \times 10^{-7}$ | $1.15 \times 10^{-3} \pm 1.23 \times 10^{-5}$ | $8.53 \times 10^{4} \pm 3.99 \times 10^{3}$ | $8.69 \times 10^{2} \pm 9.25 \times 10^{0}$ | $106.80 \pm 5.47$ |
| S60 | $4.86 \times 10^{-6} \pm 9.32 \times 10^{-7}$ | $5.28 \times 10^{-4} \pm 2.25 \times 10^{-5}$ | $2.12 \times 10^{5} \pm 3.61 \times 10^{4}$ | $1.90 \times 10^{3} \pm 8.01 \times 10^{1}$ | $117.29 \pm 4.77$ |
| S80 | $1.50 \times 10^{-6} \pm 1.34 \times 10^{-7}$ | $2.49 \times 10^{-5} \pm 1.64 \times 10^{-6}$ | $6.74 \times 10^{5} \pm 6.48 \times 10^{4}$ | $4.04 \times 10^{4} \pm 2.64 \times 10^{3}$ | $18.7 \pm 2.82$ |

In particular, for the systems with G geometry, the $R_0/R_8$ values changed from $2.21 \pm 0.20$, and in the case of the G80 structure, to $4.95 \pm 0.50$ for the less porous G40. More significantly, the systems with S geometry showed larger resistance variations with a one-order of magnitude drop when S40 structure is compared with S80 structure (Figure 7c,d). The compression sensitivity of several porous structures was evaluated by measuring the gauge factor (GF) at 8% strain (Figure 7d). All samples displayed GF absolute values above 6, with TPU/GE S40 and TPU/GE S60 displaying a value above 12. This difference is ascribed to the peculiar shape of the S unit cell, which leads to 3D structures with bigger trabeculae, which under deformation give rise to the building up of more effective conductive pathways. To the best of our knowledge, such high values have never been reported for graphene-based polymer porous structures when subjected to compressive strain [47]. It has to be pointed out that GF values are even higher for deformation extents lower than 8%, and then tend to plateau as the maximum strain value is approached (Figure 8). Indeed, most samples displayed GF absolute values ranging from 60 to 20 for deformation extents from 1% to 5%.

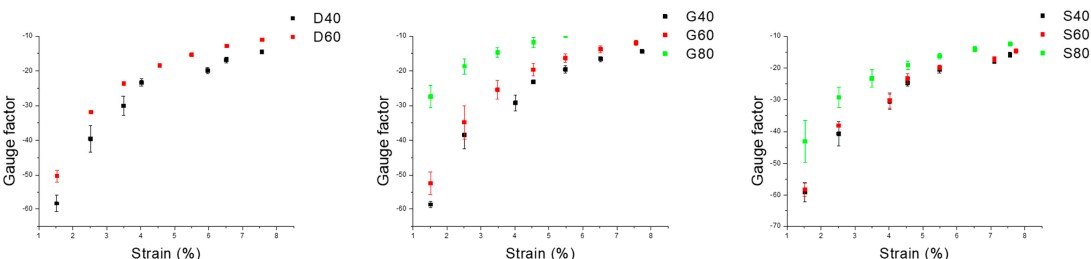

**Figure 8.** Variation of gauge factor as a function of compression strain for the TPU/GE porous structures with: (**a**) Diamond, (**b**) Gyroid, and (**c**) Schwarz unit cells.

It is also worth noting that for all porous structures, GF increases by reducing the porosity at a fixed strain. This confirms that a key role is played by the dimension of the trabeculae. The bigger the size of trabeculae, and consequently the larger the number of wrapped TPU particles which can be compacted, the larger the GF.

The outstanding sensitivity of the SLS fabricated structures demonstrates that they can be used as piezoresistors in the detection of very small deformations (i.e., strain less than 5%). All the SLS fabricated TPU/GE structures were also characterized in terms of electromechanical cycling stability. The samples were submitted to 50 consecutive compressive cycles (at 8% of strain), as reported in Figure 9a for TPU/GE S40. In the cyclic compression process, both mechanical and electrical response of the sample were stable all over the experiment, demonstrating excellent stability and signal reversibility. Figure 9b summarizes the results of the electromechanical cycling tests for all porous structures. The resistance values at 8% strain reported as a function of time clearly demonstrate that regardless of porosity and geometry, after the very first compression cycles, all structures exhibited excellent stability and repeatability.

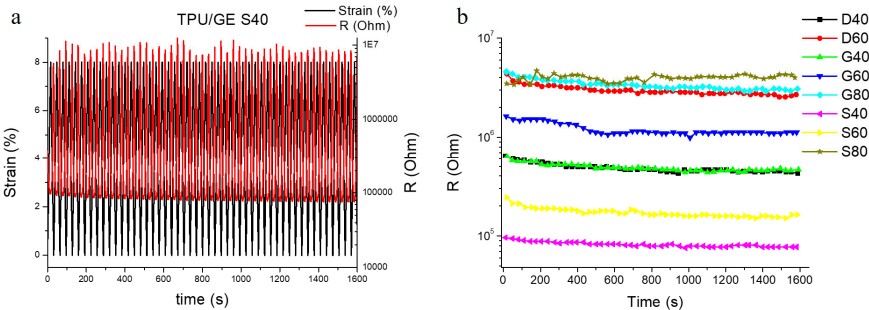

**Figure 9.** (**a**) Piezoresistive behavior of TPU/GE S40 over 50-cycle compression test, and (**b**) resistance values at 8% strain as a function of time for all TPU/GE composite structures.

### 3.4. Thermal Conductivity

The thermal conductivity ($\lambda$) of TPU/GE compact and porous structures was examined and compared with the thermal conductivity of compact TPU, to gain insight on the effect of GE addition, porosity, and geometry on their thermal behavior. Compact TPU showed a thermal conductivity of $0.24 \pm 0.012$ W(m $\times$ K) that is approximately half of the value for compact TPU/GE composites. This means that with the addition of 1% of GE, the thermal conductivity doubles its value for pristine TPU compact structures. Moreover, a porous TPU/GE structure with 40% of porosity showed values of thermal conductivity comparable with that of pristine compact TPU. Figure 10 shows that $\lambda$ is strongly affected by the porosity of the structure, making it possible to tune the thermal conductivity of the TPU/GE composites by modifying the overall porosity.

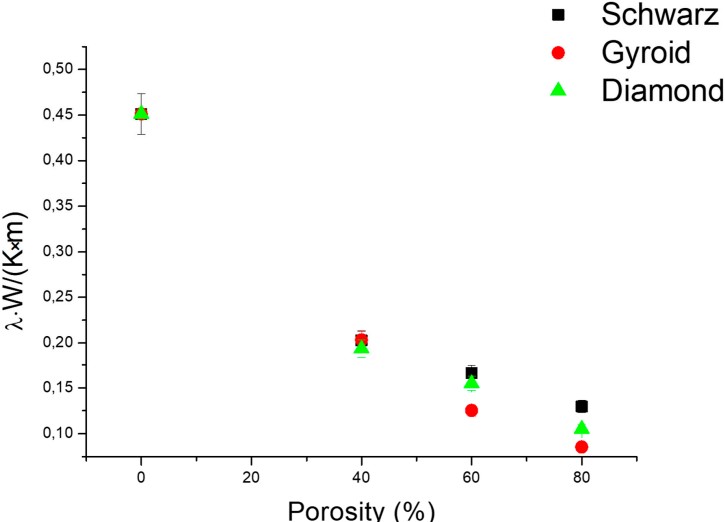

**Figure 10.** Effect of porosity and geometry on the effective thermal conductivity of TPU/GE porous structures.

Obviously, λ decreases as the porosity increases, going from 0.451 ± 0.023 W/m·K for the bulk material down to 0.086 ± 0.004 W/m·K for the Gyroid structure with 80% porosity. It was ascertained that an increase in porosity is linked to a decrease in the trabeculae size and an increase of pore dimension for a given unit cell [48]. On the other hand, results obtained for the three different architectures (D, G, and S) indicate that the effect of geometry on the thermal conductivity is small if compared with the effect on the electrical properties and piezoresistivity. This confirms that the transfer of heat phonons and electrons are subject to different physical laws. However, the thermal conductivity remained similar for the different geometries until 40% of porosity. Above this value, the Schwarz geometry showed the highest λ values, while the Gyroid displayed the lowest values, both at 60 and 80% porosity. This effect is again ascribed to the different size of trabeculae generated by the D, G, and S unit cell architectures.

## 4. Conclusions

Electrically conductive TPU/GE porous structures were successfully fabricated by SLS processing by using a home-made powder realized by wrapping GE nanoplatelets onto TPU particles. Several 3D mathematically defined architectures with different porosity extents were designed and realized, namely Gyroid, Diamond, and Schwarz. Electrical and thermal conductivity, mechanical strength, filler dispersion, and interaction with the polymer matrix of the TPU/GE porous systems were investigated and correlated with their porosity and internal architecture.

Morphological characterization clearly indicated that SLS manufacturing is suitable to create porous structures with narrow pore size distributions and high pore interconnectivity. Moreover, upon processing, the GE sheets remain entrapped in between the interparticle boundaries, thereby forming a segregated conductive network fully percolating the porous structure. GE hindered crystallization of TPU hard segments, but reduced the mobility of the polymer soft segments, increasing the $T_g$. Furthermore, GE brought about an improvement of thermal stability of TPU. Compression tests revealed that S geometry provides the porous structure with a higher elastic modulus in comparison to the corresponding D and G geometries.

All architectures showed electrical conductivity as well as negative piezoresistive behavior during cyclic compression tests, characterized by outstanding GF absolute values. In particular, S geometry structures yielded GF values of 12.4 at 8% strain, due to the combination of GE network segregation and higher size of trabeculae connecting the porosity. GF absolute values ranging from 60 to 20 were observed for deformation extents from 1% to 5%, demonstrating that the SLS-processed porous systems can be used in the detection of strains lower than 5%. Upon cyclic piezoresistive sensing tests, all samples exhibited excellent behavior repeatability, regardless of their porosity and geometry. Thermal conductivity of the TPU/GE structures significantly decreased with increasing porosity, while the effect of the structure architecture was less relevant.

The reported results demonstrate that the TPU/GE powder is a suitable material for the SLS fabrication of porous structures with highly tailored flexibility and electrical conductivity. The powder enables the obtainment of a right balance between mechanical and functional properties of the printed structures, which in turns hold great potential to be used as flexible, highly sensitive, and stable piezoresistive sensors in wearable or implantable devices, and dielectric elastomer actuators.

**Author Contributions:** Conceptualization, M.L., L.A., and H.X.; methodology and sample preparation, A.R., G.F., and X.G.; data analysis, A.R., G.R., G.G.B., and C.S.; writing—original draft preparation, A.R., G.R., and P.C.; writing—review and editing, M.L., P.C., L.A., and H.X.

**Funding:** This research was funded by Marie Skłodowska-Curie Actions (MSCA) Research and Innovation Staff Exchange (RISE) H2020-MSCA-RISE-2016, Project Acronym: Graphene 3D—Grant Number: 734164, and the National Key R&D Program of China (2017YFE01115000).

**Acknowledgments:** Thanks are due to Marianna Pannico (IPCB-CNR) for help in acquiring Raman spectra.

**Conflicts of Interest:** The authors declare no conflict of interest.

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
