# Peer review of "Selective Laser Sintering Fabricated Thermoplastic Polyurethane/Graphene Cellular Structures with Tailorable Properties and High Strain Sensitivity"

_applsci, doi:10.3390/app9050864_

Round 1
Reviewer 1 Report
The content of this study is interesting and the results are useful for the researchers in the related field. However, detailed electrical conductivity values were not provided. Therefore, detailed electrical conductivity values should be provided, and the values should be discussed about the relatioship with porosity and geometry.
Author Response
The authors are grateful to the reviewer for considering the manuscript as valuable for publications in Applied Science Journal and for his/her valuable comments that have been all considered in the manuscript revision. The modification of the manuscript, accordingly to the reviewer comments, has been highlighted in yellow into the amended manuscript.
Point 1: The content of this study is interesting and the results are useful for the researchers in the related field. However, detailed electrical conductivity values were not provided. Therefore, detailed electrical conductivity values should be provided, and the values should be discussed about the relationship with porosity and geometry
Response 1: The authors thanks the reviewer for the valuable comment; the electrical conductivity, resistivity and resistance ratio of the porous structures have been measured and the values are reported in Table 2 as a functions of morphological parameters (porosity and geometry) at low and high deformation (strain equal to 0 and 8%). All the modifications have been reported also in the manuscript at page 10 line 322.
Table 2. Electrical conductivity of TPU/GE composites as a function of porosity and internal morphology
Sample | σ (S/m) | ρ (Ω∙m) | R0/R8 | ||
0% | 8% | 0% | 8% | ||
D40 | 3.49∙10-5± 8.48∙10-6 | 1.86∙10-4±5.59 ∙10-6 | 3.03∙104±7.89∙103 | 5.38∙103± 1.67∙102 | 6.12±1.65 |
D60 | 5.01∙10-6± 1.77∙10-6 | 2.72∙10-5± 1.16∙10-6 | 2.14∙105± 4.83∙104 | 3.69∙104±2.83 ∙103 | 6.33±1.41 |
D80 | - | - | - | - | |
G40 | 3.00∙10-5± 3.42∙10-6 | 1.80∙10-4± 6.58∙10-6 | 2.52∙104± 2.32∙103 | 5.55∙103± 2.02∙102 | 4.95±0.50 |
G60 | 1.76∙10-5± 9.25∙10-7 | 7.51∙10-5± 4.80∙10-6 | 5.70∙104± 3.08∙103 | 1.34∙104± 9.59∙102 | 4.66±0.46 |
G80 | 1.32∙10-5± 3.67∙10-7 | 2.68∙10-5± 2.06∙10-6 | 7.60∙104± 2.13∙103 | 3.75∙104± 3.07∙103 | 2.21±0.20 |
S40 | 1.17∙10-5± 5.67∙10-7 | 1.15∙10-3± 1.23∙10-5 | 8.53∙104± 3.99∙103 | 8.69∙102± 9.25∙100 | 106.80±5.47 |
S60 | 4.86∙10-6± 9.32∙10-7 | 5.28∙10-4± 2.25∙10-5 | 2.12∙105± 3.61∙104 | 1.90∙103± 8.01∙101 | 117.29±4.77 |
S80 | 1.50∙10-6± 1.34∙10-7 | 2.49∙10-5± 1.64∙10-6 | 6.74∙105± 6.48∙104 | 4.04∙104± 2.64∙103 | 18.7±2.82 |
Electrical conductivity strongly depends on the percentage of porosity but also on internal architectures of the 3D structures both at low (i.e. zero deformation corresponding to the case of constant preload applied to the samples) and high deformation (i.e. 8 % deformation in compression). In details, the conductivity at zero deformation decreases as the percentage of porosity increases. In fact, considering the S geometry, σ decreased by about one order of magnitude when the porosity increased from 40 to 80%, going from 1.17·10-5 to 1.50·10-6. Similar results have been obtained when the deformation is 8%. Moreover, the piezoelectrical sensitivity varies significantly with the three geometries, as it is possible to see from the ratio of electrical resistance (R0/R8), whose values are reported in Table 2. In particular, for the systems with G geometry, the R0/R8 values changed from 2.21, in the case of the G80 structure, to 4.95 for the less porous G40. More significantly, the systems with S geometry showed larger resistance variations with a one-order of magnitude drop when S40 structure is compared with S80 structure (Figure 7c, d).

Reviewer 2 Report
The paper submitted to Applied Sciences treats about the manufacturing of cellular/porous structures of polyurethane with incorporated graphene by SLS method.
The paper is written with the proper English, easy to read and understand. The idea is nice and there are many experiment done – all necessary (maybe TGA is not required).
I suggest publishing after author follows the comments below.
1) Please doublecheck if the spot size was really 450 um (0.45 mm) and laser power 5W. I suppose that 5 W focused on such big spot would not sinter the powder.
2) Line 138 – reference error , please correct
3) The general doubt is if you can evaluate the properties of cellular structures on models having only 3x3x3cells. In general it is always good to have at least 10x10x10 cells to remove the influence of “border effects” – please comment on that.
4) Figure 4 – if there is only 1 % of “graphene” – why the signal is so strong and there is no signal on the polymer? Please comment on that.
5) Figure 5a – if you don’t have values on Y axis why you need to say they are in W/g. The measurement should be replicated and the average values with standard deviations should be given.
6) Figure 5 b – please make repetitions and show on the graph.. maybe there will be no change in general.
7) I have a very serious concern about the resistance measurement interpretation. While authors claims the observation of negative piezoresistive bahaviour I would ask : don’t you think that during the compression test the resistivity lowers due to better contact of the tested sample with the electrodes???? How do you prove it is not what really makes the resistance drop?
8) Figure 8 – don’t join the measurement points with the line- add the standard deviation markers.
9) Figure 10 – don’t join points with line and add standard deviations. There should be W/(m*K) not W/m*k (capital K and also () or W/m/K )
10) Why there is no comparison with sample with no graphene addition? Does it really improve the thermal conductivity? 0.45 is still not great conductor.
Good luck!
Author Response
The authors are grateful to the reviewer for considering the manuscript suitable for publications in Applied Science Journal and for his/her valuable comments that have been all considered in the manuscript revision. The modifications of the manuscript, accordingly to the reviewer comments, have been highlighted in yellow into the amended manuscript.
1) Please double check if the spot size was really 450 μm (0.45 mm) and laser power 5W. I suppose that 5W focused on such big spot would not sinter the powder.
Response 1: The authors thank the reviewer for his comment. Infact the authors did a mistake in reporting the spot size, which is not available for the SLS model 3D printer used in this work. Thus, the spot size of the laser has been removed from the paragraph. Moreover, as reported in Table 1, the laser power of the machine was 60W, while the output power was set to only 5W to prevent the printed sample to sticking to the powder and causing a decrease in the printing accuracy. This consideration has been reported in the main text at pag. 4 line142.
2) Line 138 – reference error, please correct.
Response 2: The authors thank the reviewer for the comment; the reference has been added to the text. (Pag 3 line 133)
3) The general doubt is if you can evaluate the properties of cellular structures on models having only 3x3x3cells. In general, it is always good to have at least 10x10x10 cells to remove the influence of “border effects” – please comment on that.
Response 3: The authors thank the reviewer for the valuable comment. The authors decide to build models having 3x3x3 unit cells because of by increasing the number of unit cells that would have brought to a decrease of the thickness of the internal trabeculae (as reported in Figure 1 below). If the trabeculae are too thin, the laser is not able to sinter the TPU particles resulting in structure with no coherence.
Figure 1: trabecular thickness as a function of porosity and number of unit cells (figure is reported in the attached pdf file).
Moreover, if we use the same unit cell of 3x3x3 structures and we repeat it 10 times as suggested we obtain structures measure 30x30x30 mm3 mm that is too big for application as wearable sensor.
4) Figure 4 – if there is only 1 % of “graphene” – why the signal is so strong and there is no signal on the polymer? Please comment on that.
Response 4: The signal is so strong because the analysis has been conducted by using a Raman microscopy which allows to selectively focus onto the TPU particles or onto the segregated graphene between particles. Additional comment has been added to the manuscript at pag. 7 line 228.
5) Figure 5a – if you don’t have values on Y axis why you need to say they are in W/g. The measurement should be replicated and the average values with standard deviations should be given.
Response 5: The authors thank the reviewer for the valuable comment; Figure 5a has been modified accordingly with the reviewer suggestions and the average value with standard deviation of melting and crystallization temperatures have been reported into the manuscript at pag. 8.
6) Figure 5b – please make repetitions and show on the graph. Maybe there will be no change in general.
Response 6: The TGA analysis of TPU/GE nanocomposites has been repeated 5 times and the results have been reported in the Figure 2 below. As expected the TPU/GE has always the same trend and it is totally independent from the geometry of the 3D structure. For this reason, in the manuscript we reported just one curve for TPU and one for TPU/GE highlighting the influence of GE on the thermal degradation of the material. Moreover, mean value and standard deviation of degradation temperature have been reported into the manuscript at pag 8.
Figure 2: TGA curves for TPU and TPU/GE nanocomposites samples from different geometries (figures is reported in the attached pdf file).
7) I have a very serious concern about the resistance measurement interpretation. While authors claim the observation of negative piezoresistive behaviour I would ask: don’t you think that during the compression test the resistivity lowers due to better contact of the tested sample with the electrodes? How do you prove it is not what really makes the resistance drop?
Response 7: The authors thank the reviewer for the valuable comments. Before measurement, the porous structures were pre-compressed to a strain value of 4% in order to ensure the contact between the electrode and the samples (as reported in the manuscript at pag. 4 line 175)
8) Figure 8 – don’t join the measurement points with the line- add the standard deviation markers.
Response 8: The authors would like to thank the reviewer for the comment; In the revised version of the manuscript, Figure 8 has been modified accordingly with the review suggestion. The line has been removed and the standard deviation marker has been added.
9) Figure 10 – don’t join points with line and add standard deviations. There should be W/(m*K) not W/m*k (capital K and also () or W/m/K)
Response 9: The authors thank the reviewer for the comment; In the revised version, Figure 10 has been changed in agreement with the suggestion of the reviewer, the line has been removed and also the label of Y axis has been modified.
10) Why there is no comparison with sample with no graphene addition? Does it really improve the thermal conductivity? 0.45 is still not great conductor.
Response 10: The authors would like to thank the reviewer for the comment. The thermal conductivity of the compact TPU has been measured and the value is 0.24 W/(m·K). This means that by adding 1wt% of GE the thermal conductivity of porous systems doubles its value to 0.45 W/(m·K). The comparison between thermal conductivity of pristine compact TPU and graphene-based porous systems has been reported in the manuscript at pag. 12 line 368.
